# Retrospective Multicenter Study on Outcome Measurement for Dyskinesia Improvement in Parkinson’s Disease Patients with Pallidal and Subthalamic Nucleus Deep Brain Stimulation

**DOI:** 10.3390/brainsci12081054

**Published:** 2022-08-09

**Authors:** Fangang Meng, Shanshan Cen, Zhiqiang Yi, Weiguo Li, Guoen Cai, Feng Wang, Stephan S. Quintin, Grace E. Hey, Jairo S. Hernandez, Chunlei Han, Shiying Fan, Yuan Gao, Zimu Song, Junfei Yi, Kailiang Wang, Liangwen Zhang, Adolfo Ramirez-Zamora, Jianguo Zhang

**Affiliations:** 1Beijing Neurosurgical Institute, Capital Medical University, Beijing 100070, China; 2Department of Functional Neurosurgery, Beijing Tiantan Hospital, Capital Medical University, Beijing 100070, China; 3Beijing Key Laboratory of Neurostimulation, Beijing 100070, China; 4Chinese Institute for Brain Research, Beijing 102206, China; 5Department of Neurology, Xuanwu Hospital, Beijing 100053, China; 6Department of Neurosurgery, Peking University First Hospital, Beijing 100034, China; 7Department of Neurosurgery, QiLu Hospital of Shandong University, Jinan 250012, China; 8Department of Neurology, Fujian Medical University Union Hospital, Fuzhou 350001, China; 9Department of Neurosurgery, General Hospital of Ningxia Medical University, Yinchuan 750004, China; 10Departments of Neurosurgery, The First Affiliated Hospital, Zhejiang University School of Medicine, Hangzhou 310058, China; 11Department of Neurology, Fixel Center for Neurological Diseases, Program in Movement Disorders and Neurorestoration, University of Florida, Gainesville, FL 32607, USA; 12Department of Neurosurgery, Shandong Provincial Hospital Affiliated to Shandong University, Jinan 250021, China

**Keywords:** Parkinson’s disease, levodopa-induced dyskinesia, deep brain stimulation, Unified Dyskinesia Rating Scale, levodopa equivalent daily dose

## Abstract

Deep brain stimulation (DBS) is an effective treatment for dyskinesia in patients with Parkinson’s disease (PD), among which the therapeutic targets commonly used include the subthalamic nucleus (STN) and the globus pallidus internus (GPi). Levodopa-induced dyskinesia (LID) is one of the common motor complications arising in PD patients on chronic treatment with levodopa. In this article, we retrospectively evaluated the outcomes of LID with the Unified Dyskinesia Rating Scale (UDysRS) in patients who underwent DBS in multiple centers with a GPi or an STN target. Meanwhile, the Med off MDS-Unified Parkinson’s Disease Rating Scale (MDS-UPDRS-Ⅲ) and the levodopa equivalent daily dose (LEDD) were also observed as secondary indicators. PD patients with a GPi target showed a more significant improvement in the UDysRS compared with an STN target (92.9 ± 16.7% vs. 66.0 ± 33.6%, *p <* 0.0001). Both the GPi and the STN showed similar improvement in Med off UPDRS-III scores (49.8 ± 22.6% vs. 52.3 ± 29.5%, *p =* 0.5458). However, the LEDD was obviously reduced with the STN target compared with the GPi target (44.6 ± 28.1% vs. 12.2 ± 45.8%, *p =* 0.006).

## 1. Introduction

In 1817, James Parkinson described a progressive neurological disease characterized by tremors, compulsion, and slowness of movements. Charcot later referred to the disease as Parkinson’s disease (PD) in the 19th century, also along with a spectrum of non-motor symptoms including mood, sleep, and cognitive symptoms [1]. Levodopa-induced dyskinesia (LID) is one of the common motor complications arising in patients with PD on chronic treatment with levodopa, which refers to involuntary movements other than tremors and most commonly consists of chorea [2,3]. As one of the side effects of dopaminergic therapy, LID represents a major clinical problem in the management of patients with PD that has influenced treatment strategies. It was later recognized that LID can also be caused by dopamine agonists and not just levodopa treatment [4]. LID usually occurs with advancing disease, with a high risk especially in younger PD patients treated with higher doses and long-term therapy with levodopa, as PD patients begin to experience “cycles” of motor dysfunction. Approximately 40% of PD patients may experience LID after 4 years, and 50% to 80% of PD patients treated with levodopa for more than 5 to 10 years may experience LID [2,3,5,6]. LID reduces the required dose of dopaminergic agents for symptom control during peak-dose dyskinesia [7]. Compared with peak-dose dyskinesia, diphasic dyskinesia tends to have a more variable topographical distribution and be more troublesome [3]. LID is a common and significant cause of disability in PD and a major indication for surgical/advanced treatments [8].

The use of deep brain stimulation (DBS) for treating depression has been proven to be an effective and safe alternative. There is good evidence that DBS treatment can be applied to both the subthalamic nucleus (STN) and the globus pallidus internus (GPi) [6,9]. A large number of studies, including randomized controlled trials, have shown that this approach can significantly improve key symptoms of PD, such as tremors, rigidity, and bradykinesia [9,10,11,12,13]. A number of studies have found that the STN and the GPi are effective in controlling dyskinesia. There have been several studies comparing GPi- and STN-DBS in terms of typical PD symptoms, but relatively few have compared the two for the control of dyskinesia and long-term results focused on LID [4,14]. Studies on patients with Parkinson’s disease have not shown agreement on the most effective stimulation target for controlling dyskinesia. In the past, it has been unknown what mechanisms underlie the reduction in dyskinesia after DBS. It is generally recommended to reduce medication, with a relatively greater reduction in the levodopa equivalent daily dose (LEDD) for the management of dyskinesia with STN-DBS [6,15,16]. In comparison to STN-DBS, GPi-DBS offers a marginal reduction in the dopaminergic dose; however, it may directly affect dyskinesia [17]. Despite this, we still do not completely understand the underlying mechanisms.

Up to now, different scales have been used in DBS studies without a standardized, consistent assessment. Generally, the outcome of the dyskinesia change was assessed using the Unified Dyskinesia Rating Scale (UDysRS), which can simultaneously assess some kinds of objective and subjective symptoms of dyskinesia [17]. It is a well-established tool and a fully validated scale for neurologists to assess the severity and the disability associated with dyskinesia in patients with PD. Since dyskinesia is important, and there is little literature surrounding it, we conducted a retrospective study in patients with preoperative dyskinesia in our center. After DBS treatment, we evaluated dyskinesia severity and presence using the Unified Dyskinesia Rating Scale (UDysRS). In this study, we examined whether STN- or GPi-DBS was more effective in treating LID, as well as exploring possible mechanisms associated with the relief of dyskinesia following DBS.

## 2. Methods

### 2.1. Participants

In this study, we retrospectively evaluated the outcomes in clinics of preoperative LID in PD patients undergoing DBS, which targeted the STN or the GPi, at five medical centers including Beijing Tiantan Hospital, Capital Medical University; Peking University First Hospital; the General Hospital of Ningxia Medical University; the Qilu Hospital of Shandong University; and the Fujian Medical University Union Hospital, from August 2015 to June in 2019. Patients diagnosed with advanced idiopathic PD on the basis of the International Parkinson and Movement Disorder Society (MDS) criteria and with LID prior to surgery were screened. At the same time, we also excluded several neurological conditions that mimic the advanced form of PD, including dystonia and other movement disorders [18,19]. Finally, a total of 61 cases were confirmed. Patients with a previous thalamectomy or pallidotomy and with a failure of follow-up were precluded. Finally, 56 patients were included in this research, including 32 patients who underwent STN-DBS and 24 patients who underwent GPi-DBS, as shown in Table 1. The stimulation targets were decided randomly. Bilateral STN or GPi stimulation had been used for all the patients. In addition, all of the patients had signed informed consent forms. At the same time, the research was ratified by the ethics committee of Beijing Tiantan Hospital (KY2021-159-01).

### 2.2. Surgical Procedure

DBS electrodes were implanted under local anesthesia employing the Leksell stereotactic system (Elekta Instrument AB, Sweden). A recording of a single unit of the intraoperative procedure and a high frequency stimulation test were used to determine the best location for permanent electrode implantation. Quadripolar electrodes (3389 for the STN target, 3387 for the GPi target; Medtronic, Minneapolis, MN, USA or L301 for the STN target, L302 for the GPi target; PINS Medical, Beijing, China) were implanted in all the patients. The initial stereotactic STN coordinates were posterior to the mid commissural point (MCP) by 2–3 mm, under the intercommissural line for the lower contact by 4–6 mm, and lateral to the anterior commissure–posterior commissure (AC–PC) by 12–14 mm. The GPi target coordinates were located 2 mm in front of the MCP, 6–9 mm under the intercommissural line of the lower contact, and 18–22 mm anterolateral to the AC–PC. Intraoperative single recordings were used during the determination of the location of the STN and the GPi. Under general anesthesia, electrodes were attached to the implantable pulse generator (IPG), which was inserted in the subclavicular region following lead implantation. Following surgery, postoperative CT was used to exclude acute intracranial complications and MRI was used to confirm the electrode placement. After surgery, each patient received routine adjustments to the stimulation setting and medication until symptoms were under optimal control. In addition, the IPG was turned on one month after DBS surgery. Normally, the improvement of the symptoms will reach its stabilization about six months after the surgery. In addition, patients scheduled follow-up visits for evaluation in clinic and the adjustment of stimulus settings and medication at least six months afterwards. All the postoperative DBS parameter settings were adjusted when the subjects were discontinued from taking any dopaminergic medication for at least 12 h.

### 2.3. Assessments

Assessments were performed one to two weeks before surgery and then every six months after stimulation. The UDysRS entries were grouped into four subfields: impairment, disability, on-dyskinesia, and off-dystonia, of which the last two were objective and the first two sections were historical. Motor evaluations were assessed by using the MDS-UPDRS-Ⅲ. The LEDD was calculated according to the formula in the literature. For example, 100 mg of standard L-dopa is equivalent to 130 mg of controlled-release L-dopa, 10 mg bromocriptine, 1 mg pergolide, 1 mg apomorphine, 1 mg lisuride, 6 mg ropinirole, and 60 mg piribedil [20]. The improvement in clinic was calculated as the percentage of the result of (Pre-scores − Pos-scores)/Pre-scores.

### 2.4. Statistical Analyses

The descriptive analysis method was used to analyze the demographic information, describing it as the mean ± standard deviation (SD) or the median of the quartile interval. An Independent Samples T test and a Mann–Whitney U test were performed to compare the age of onset, the course of the disease, LID, the LEDD, and so on. In addition, a Chi-squared test was performed to contrast the groups by sex. A nonpaired samples T test and a nonparametric paired samples Wilcoxon signed rank test were used to compare the UDysRS scores and the Med off MDS-UPDRS-III scores at baseline and at the last follow-up. The threshold of the statistical data was confirmed at *p* < 0.05. IBM SPSS (version 25) software was used for statistical analysis.

## 3. Results

### 3.1. Characteristics of Patients

The demographics of the 32 patients with STN-DBS and the 24 patients with GPi-DBS are summarized in Table 1. In addition, the cohort’s median follow-up was 24 months for GPi-DBS and 18 months for STN-DBS. The two groups of patients resembled one another with regard to the age of onset (*p* = 0.769), gender (*p* = 0.171), the duration of disease at DBS (*p* = 0.142), the age at DBS (*p* = 0.268), the LEDD (*p* = 0.393), LID (*p* = 0.056), the Med off MDS-UPDRS-III score (*p* = 0.118), and the time of follow-up (*p* = 0.291).

### 3.2. Outcome of UDysRS Scale

The effects of STN- and GPi-DBS on the UDysRS score are described in Figure 1A,B. The UDysRS scores in patients with STN- and GPi-DBS were significantly lower than the baseline at the last follow-up. The Unified Dyskinesia Rating Scale (UDysRS) score with STN-DBS was 66.0 ± 33.6% and with GPi-DBS, it was 92.9 ± 16.7%. The GPi patients showed a more significant improvement in LID compared with patients with STN-DBS (*p <* 0.0001).

### 3.3. Outcome of the Med Off MDS-UPDRS-III

The Med off MDS-UPDRS-III scores after STN- and GPi-DBS were significantly improved compared to the baseline at the last follow-up. The Med off MDS-UPDRS-III median improvement at the last follow-up was 52.3 ± 29.5% and 49.8 ± 22.6%, respectively, for patients with STN- and GPi-DBS, which can be seen in Figure 1C,D. However, the difference between STN-DBS and GPi-DBS in the MDS-UPDRS-III improvement was not prominent (*p =* 0.5458).

### 3.4. Outcome of the Levodopa Equivalent Daily Dose (LEDD)

In the STN group, the LEDD was obviously reduced from 871.6 ± 364.1 mg to 444.2 ± 277.2 mg; the reduction rate was 44.6 ± 28.1%. In the GPi group, the LEDD was reduced from 973.9 ± 418.5 mg to 750.4 ± 338.7 mg; the reduction rate was 12.2 ± 45.8%. Compared with the GPi group, the STN group showed an obvious reduction (*p =* 0.006).

### 3.5. A Grouping Approach for Dyskinesia Based on the UDysRS

Based on the UDysRS, we can group the results for dyskinesia as a supplement. Group I means dyskinesia almost disappears, with a 76–100% reduction in the UDysRS scores, IA means no dyskinesia, and IB means a little dyskinesia is present but it has no effect on the patients; Group II means dyskinesia improves significantly, with a 51–75% reduction in the UDysRS scores; Group III means dyskinesia improves partially, with a 26–50% reduction in the UDysRS scores; Group IV means dyskinesia improves slightly, with a 0–25% reduction in the UDysRS scores, IVA means not improved at all, and IVB means a little change, but it has no benefit to the patients; Group V means dyskinesia is aggravated, UDysRS scores are increased, VA means the original dyskinesia symptoms are aggravated, and VB means new dyskinesia symptoms appear and do not alleviate. In summary, group I–IV indicate a 76–100%, 51–75%, 26–50%, and 0–25% level of dyskinesia improvement, respectively, whereas group V indicates the worsening of dyskinesia.

In this cases series, 16/32 of the STN-DBS group had a Group I level of improvement of LID, 8/32 of the STN-DBS group reported a Group II level of improvement, 5/32 and 1/32 of the STN-DBS group had Group III and IV levels of improvement of LID; however, there were 2/32 of the STN-DBS group who reported worsening of LID (Group V) at the last follow-up. Compared to the STN-DBS group, all of the GPi-DBS group showed Group I–III levels of improvement of LID (21/24, 2/24, and 1/24, respectively), and none of the GPi-DBS group reported worsening of LID (0/24), as shown in Table 2.

## 4. Discussion

DBS has been reported to be used to treat several neurobehavioral disorders such as chronic pain, major depression, Tourette’s Syndrome, addiction, obsessive–compulsive disorder and epilepsy, etc., and is mainly applied to patients with PD symptoms. Although, up to now, its potential molecular and physiological mechanism is still unknown, it has been reported that DBS may act via a few mechanisms including the electrical and neurochemical effects of stimulation in the network, the modulation of oscillatory activity and synaptic plasticity, and, potentially, neuroprotection and neurogenesis, either directly, or by modulating the circuitry or signaling pathways [21,22,23,24,25,26,27,28]. LID remains a significant problem in patients with PD, which has a significant negative impact on health-related quality of life and health economic outcomes [29]. It is generally believed that DBS of the GPi has a direct and profound antidyskinetic effect, whereas the relief of dyskinesias by DBS of the STN depends on a postoperative reduction in dopaminergic medications, plastic changes of the basal ganglia circuits that modulate L-dopa responsiveness, and stimulation of pallidothalamic fibers in the zona incerta [3,15]. In addition, research in recent years has shown that high-frequency electrical stimulation of the area above the STN can directly suppress levodopa-induced on-dyskinesia [30]. We discovered that GPi-DBS demonstrated a better clinical outcome for LID compared to STN-DBS in this multicenter retrospective analysis (*p* < 0.0001), although GPi- and STN-DBS showed similar improvement in Med off MDS-UPDRS-III scores (*p* = 0.5458).

The UDysRS has been developed as a comprehensive rating tool of dyskinesia in PD, which has been used worldwide for many years [31]. Consistent with the goals of PD/LID treatment studies, the improvement scale is a dependable tool for detecting changes in the seriousness of dyskinesias and can be helpful to determine therapeutic indicators for experimental medicine or devices [17]. Scales that precisely measure differences are useful for the comprehensive evaluation of drug efficacy and treatment windows for translational preclinical studies.

There have been few studies comparing GPi- and STN-DBS for the control of dyskinesia and long-term results focused on LID [4,14]. It is generally recommended to reduce medication with a relatively greater reduction in the levodopa equivalent daily dose (LEDD) for the management of dyskinesia with STN-DBS [6,15,16]. In 2017, Juhász A et al. found that GPi- DBS offers a marginal reduction in the dopaminergic dose; however, it may directly affect dyskinesia in contrast to STN-DBS [17]. In this study, we grouped the UDysRS scale for PD patients with dyskinesia. It can be recognized as a supplement for the UDysRS scale. Group I means dyskinesia almost disappears, with a 76–100% reduction in UDysRS scores; Group II means dyskinesia improves significantly; Group III means dyskinesia improves partially; Group IV means dyskinesia improves slightly; and Group V means dyskinesia is aggravated and UDysRS scores increase. In summary, group I-IV indicate a 76–100%, 51–75%, 26–50%, and 0–25% dyskinesia level of improvement, respectively, whereas group V indicates the worsening of dyskinesia. Scale of improvement grades or groups can also provide helpful information about level changes in dyskinesias.

Because GPi-DBS showed a similar improvement in Med off UPDRS-III scores, a better outcome for LID, and a disadvantage in terms of drug reduction compared with STN-DBS, it can be concluded that GPi-DBS might be preferred for PD patients with significant dyskinesia who do not take a large LEDD. Meanwhile, STN-DBS showed better results for drug reduction and a disadvantage in LID control; therefore, it could be considered for use for PD patients who take a large LEDD but with less serious dyskinesia. In other words, for PD patients who want to reduce LID, the GPi might be a preferred target; for patients who want to reduce their dosage of the drug, STN could be considered.

There are a few limitations in our study. As a first point, our study was a retrospective one conducted in China. Ideally, clinical DBS specialists all over the world could perform a randomized controlled trial in the near future to confirm our study results. Secondly, DBS management of PD patients with LID is complicated, and other factors including cognitive evaluation, psychiatric problems, depression and anxiety, etc., also need to be considered. Thirdly, the statistical analysis of this research is insufficiently rigorous, and the data of the scales were assessed by specialists from different centers and may be biased, so there is a certain statistical bias in the final results. Nevertheless, this is the largest study in China, and we believe that it would be helpful to further understand the current situation of DBS in China.

## 5. Conclusions

In conclusion, GPi-DBS and STN-DBS showed a similar improvement in Med off MDS-UPDRS-III scores. Compared to the STN target, GPi-DBS showed a better clinical outcome for preoperative LID based on the UDysRS, whereas STN-DBS showed an obvious reduction in the LEDD.

## Figures and Tables

**Figure 1 brainsci-12-01054-f001:**
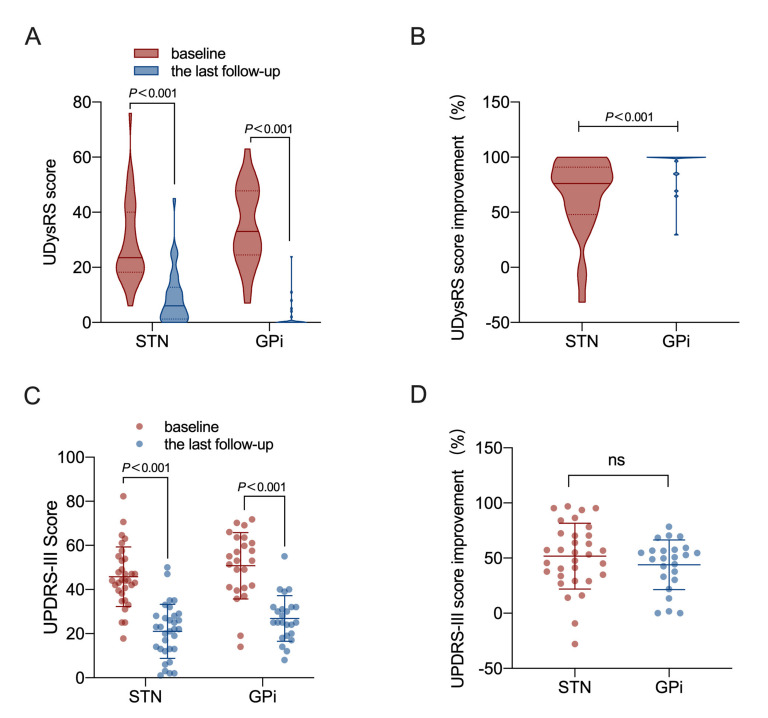
The STN-DBS and GPi-DBS influence on the Med off MDS-UPDRS-III score and UDysRS score in 56 patients who have LID. (**A**) UDysRS scores of GPi-DBS and STN-DBS patients at the last follow-up and baseline. (**B**) Contrast of the improvement of UDysRS score by GPi-DBS and STN-DBS. (**C**) Med off MDS-UPDRS-III scores of GPi-DBS and STN-DBS patients at the last follow-up and baseline. (**D**) Contrast of the improvement of Med off MDS-UPDRS-III score by GPi-DBS and STN-DBS. UDysRS, Unified Dyskinesia Rating Scale; DBS, deep brain stimulation; STN, subthalamic nucleus; GPi, globus pallidus interna; MDS-UPDRS-III, Unified Parkinson’s Disease Rating Scale part III; ns, non-significant.

**Table 1 brainsci-12-01054-t001:** The information in clinic and baseline demographics of all patients.

Variable	STN	GPi	*p* Value
Number	32	24	
Gender (M/F)	9/23	11/13	0.171
Age of onset (year)	47.8 ± 10.7	47.9 ± 9.5	0.769
Duration of disease at DBS (year)	9.5 ± 7.3	12.1 ± 3.7	0.142
Age at DBS (year)	59.3 ± 9.9	60.0 ± 8.5	0.268
Levodopa equivalent daily dose (mg/day)	871.6 ± 364.1	973.9 ± 418.5	0.393
Dyskinesia score (UDysRS)	23.5 (18.3–40.0)	33.0 (24.5–47.8)	0.056
Med off MDS-UPDRS-III score	45.8 ± 13.5	53.0 ± 15.0	0.118
Hoehn–Yahr stage			
2	5	2	
2.5	7	6	
3	16	13	
4	4	3	
Follow-up time (month)	18 (6.0–34.5)	24 (18.0–30.0)	0.291

Values are expressed as means ± standard deviation (SD) or median (Interquartile Range). *p* values are used to compare between groups based on analysis of Independent Samples T test, Mann–Whitney U test, and Chi-Squared test (sex). MDS-UPDRS-III, MDS-Unified Parkinson’s Disease Rating Scale III; GPi, globus pallidus interna; UDysRS, Unified Dyskinesia Rating Scale; DBS, deep brain stimulation; STN, subthalamic nucleus.

**Table 2 brainsci-12-01054-t002:** A grouping approach for Dyskinesia based on the UDysRS.

Group	Improvement	STN-DBS, No. (%)	GPi-DBS, No. (%)
I IA IB	Dyskinesia almost disappears, 76–100% reduction in UDysRS scores No dyskinesia A little dyskinesia present but has no effect on patients	16 (50%) 6 (18.8%) 10 (31.2%)	21 (87.5%) 18 (75%) 3 (12.5%)
II	Dyskinesia improved significantly, 51–75% reduction in UDysRS scores	8 (25%)	2 (8.3%)
III	Dyskinesia improved partially, 26–50% reduction in UDysRS scores	5 (15.6%)	1 (4.2%)
IV IVA IVB	Dyskinesia improved slightly, 0–25% reduction in UDysRS scores Not improved at all A little change, but has no benefit to patients	1 (3.1%) 1 (3.1%) 0 (0%)	0 (0%) 0 (0%) 0 (0%)
V VA VB	Dyskinesia is aggravated, increase in UDysRS scores The original dyskinesia symptoms are aggravated New dyskinesia symptoms appear and do not alleviate	2 (6.3%) 1 (3.1%) 1 (3.1%)	0 (0%) 0 (0%) 0 (0%)

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
