# Peer review of "Retrospective Multicenter Study on Outcome Measurement for Dyskinesia Improvement in Parkinson’s Disease Patients with Pallidal and Subthalamic Nucleus Deep Brain Stimulation"

_brainsci, 2022, doi:10.3390/brainsci12081054_

Round 1

Reviewer 1 Report

Major

This study does not appear to have compelling novelty compared to prior study from this group (Pallidal versus subthalamic nucleus deep brain stimulation for levodopa‐induced dyskinesia)

The authors observed that reduction in UDysRS was enhanced after GPi DBS, but that LEDD reduction was enhanced after STN DBS. The authors conclude that "GPi should be preferred for PD patients with significant dyskinesia who do not take a large LEDD" whereas "STN... should be preferred for PD patients who take a large LEDD but without so serious dyskinesia." This conclusion is not supported by the data. First, it must be clarified whether UDysRS measured in the ON or OFF state. More broadly put, did STN patients show less UDysRS improvement when controlling for LEDD change, and did GPi show less LEDD reduction when controlling for UDysRS change. 

Minor

Fig1A/B: Can the author explain why the width for GPi follow-up data is so diminutive.

Ln 199: There is no Figure 2.

There are several other minor grammatical & spelling errors that should be fixed.

Author Response

1.We agree with the reviewers that similar topic has been published previously. Nevertheless, the previously study focused on one DBS center, while our study retrospectively evaluated the outcomes in clinic of LID in patients undergoing DBS, which targeted for the STN or GPi at 5 medical centers.

2. Firstly, both of GPi and STN showed similar improvement in Med off UPDRS-III scores (49.8±22.6% vs 52.3±29.5%, P=0.5458). Secondly, LEDD was obvious reduce in STN compared with GPi (44.6±28.1% vs 12.2±45.8%, P=0.006), but UDysRS improvement in GPi is better than STN (92.9±16.7% vs 66.0±33.6, P<0.0001). So we can conclude that STN should be preferred for PD patients who take a large LEDD but without so serious dyskinesia, whereas GPi should be preferred for PD patients with significant dyskinesia who do not take a large LEDD.

3. Because UDysRS scores and UDysRS score improvements of patients in the GPi group were relatively concentrated.

4. We are sorry for that. Actually, the “figure 2” was corrected as “table 2”.

5. We have corrected all grammatical & spelling errors throughout the manuscript.

Reviewer 2 Report

The research manuscript entitled "Retrospective Multicenter Study on Outcome Measurement for Dyskinesia Improvement in Parkinson’s Disease Patinets with Pallidal and Subthalamic Nucleus Deep Brain Stimulation" by Meng et al is well and logically written. However, to improve the manuscript to a level of this journal and for the general and specialized audience below suggestion can be considered

1) there is a spelling error or typo in the title. change "Patinets" to "patients"

2) as this is a retrospective study and not a prospective study,  authors must indicate the limitations and caveats so that results can be concluded conservatively.

3) Literature coverage is not deep. please include historic background  of DBS with that of the latest literature.  DBS has a well-known effect and/or the potential to treat several neuro-behavioral conditions such as chronic pain, major depression,  obsessive-compulsive disorder (OCD), epilepsy etc (please add more ), and is mainly and well known for Parkinson's disease symptoms. However, the molecular and physiological underpinning of its effect is still not well investigated. e,g DBS upregulates genes involved in synaptic function, cell survival, and neurogenesis directly or by modulating circuitry or signaling pathways (PMID: 29570050, 26510756,  26539102, 26539102, 34936033, 20460439, 34552196, 21455927, 28713248). Authors are encouraged to add and expand to discuss this in the context of their results. 

4) Figures and tables are well prepared. 

Author Response

  1. Thank you, we corrected this spelling error in the title.
  2. Thank you, we have added the limitations and caveats in the end of this manuscript.
  3. Thank you so much for all insightful comments and suggestions. We have added background and mechanism into the discussion along with references.
  4. Thank you for the positive comments.

Reviewer 3 Report

The authors reported a study about outcome measurement for dyskinesia in PD patients with DBS. I have some comments to the authors:  

 - There is a typo in the title “patients”  

- First line of the introduction, just state that PD is a neurodegenerative disorder, describe the key motor features and also add the non-motors symptoms that can be associated.  

- Please expand the exclusion criteria of your study. The advanced form of PD is often a mimic for several neurological condition. I recommend to the authors these references to be added, because they are necessary for the readers in order to be sure that you have excluded all the possible conditions (i.e. functional neurological disorders; dystonia; and other movement disorders):  

Gorodetsky et al. Functional Patients Referred for Deep Brain Stimulation: How Common Is it? Movement Disorders Clinical Practice 2022  

Tinazzi M, et al. Demographic and clinical determinants of neck pain in idiopathic cervical dystonia. J Neural Transm (Vienna). 2020

 Ali K, Morris HR. Parkinson's disease: chameleons and mimics. Pract Neurol. 2015 Feb;15(1):14-25. doi: 10.1136/practneurol-2014-000849. Epub 2014  

- In the Table 1 HY 3.5 does not exist, please include the patient in HY 3 o 4.

 - Figure 2 is missing (you probably mean Table 2)  

- Please specify whether you have followed previous studies for the grouping approach. Your statistical analysis is just the comparison between the two group of patients. I there any statistical difference using your grouping approach? Did you analysis your findings with a regression model?

 - Discussion: i) compare your findings with the literature; ii) include a paragraph with the limitations; iii) conclusions are too strong for a retrospective study with 56 patients.               

Author Response

  1. Thank you, we corrected this spelling error in the title.
  2. Thank you, we have added the non-motors symptoms into the introduction.
  3. Thank you so much for all insightful comments and suggestions. We have checked the exclusion criteria and excluded all the possible conditions.
  4. Thank you, we corrected this HY of the patients in table 1.
  5. Thank you, we corrected this spelling error in the title one by one carefully.
  6. In this research, we have followed previous studies for the grouping approach by different targets. PD patients with GPi target showed a more significant improvement in UDysRS compared with STN (92.9±16.7% vs 66.0±33.6%, P<0.0001). Both of GPi and STN showed similar improvement in Med off UPDRS-III scores (49.8±22.6% vs 52.3±29.5%, P=0.5458). However, LEDD was obvious reduce in STN compared with GPi (44.6±28.1% vs 12.2±45.8%, P=0.006). And we didn’t analysis the findings with a regression model.

Round 2

Reviewer 3 Report

I would like to the authors for the efforts in improving the paper. However, they missed to check some points that I have raised:

1) First line of the introduction, just state that PD is a neurodegenerative disorder. Stating that PD is a kind of parkinsonism with PD-like symptoms is not clear.

2) You should state in the methods whether you excluded these type of patients. I do think this point would be beneficial for the readers. Indeed, as I mentioned in my first report, he advanced form of PD is often a mimic for several neurological condition. I recommend to the authors these references to be added, because they are necessary for the readers in order to be sure that you have excluded all the possible conditions (i.e. functional neurological disorders; dystonia; and other movement disorders):  

Gorodetsky et al. Functional Patients Referred for Deep Brain Stimulation: How Common Is it? Movement Disorders Clinical Practice 2022  

Tinazzi M, et al. Demographic and clinical determinants of neck pain in idiopathic cervical dystonia. J Neural Transm (Vienna). 2020

 Ali K, Morris HR. Parkinson's disease: chameleons and mimics. Pract Neurol. 2015 Feb;15(1):14-25. doi: 10.1136/practneurol-2014-000849. Epub 2014  

3) The statistical analysis is still too weak, please try to improve the power of your study, or state it in the limitations.

4) Comparison with the present data in the literature should be expanded also in the discussion, including all the necessary references.
